# Business as Usual with Article Processing Charges in the Transition towards OA Publishing: A Case Study Based on Elsevier

**Sergio Copiello** 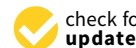

Department of Architecture, IUAV University of Venice, Dorsoduro 2206, 30123 Venice, Italy;
sergio.copiello@iuav.it; Tel.: +39-041-257-1387

**Abstract:** This paper addresses the topic of the article processing charges (APCs) that are paid when publishing articles using the open access (OA) option. Building on the Elsevier OA price list, company balance sheet figures, and ScienceDirect data, tentative answers to three questions are outlined using a Monte Carlo approach to deal with the uncertainty inherent in the inputs. The first question refers to the level of APCs from the market perspective, under the hypothesis that all the articles published in Elsevier journals exploit the OA model so that the subscription to ScienceDirect becomes worthless. The second question is how much Elsevier should charge for publishing all the articles under the OA model, assuming the profit margin reduces and adheres to the market benchmark. The third issue is how many articles would have to be accepted, in an OA-only publishing landscape, so that the publisher benefits from the same revenue and profit margin as in the recent past. The results point to high APCs, nearly twice the current level, being required to preserve the publisher's profit margin. Otherwise, by relaxing that constraint, a downward shift of APCs can be expected so they would tend to get close to current values. Accordingly, the article acceptance rate could be likely to grow from 26–27% to about 35–55%.

**Keywords:** publishing industry; for-profit and non-profit publishers; Open Access; subscription-based journals; Article Processing Charges

## 1. Introduction

The academic publishing industry has recently seen the conclusion of a dispute between Elsevier management and the editorial board members of the Journal of Informetrics (JOI) [1,2], which led those members to resign from their positions in early January 2019. That dispute resembled the 2006 case of Topology [3]—also run by Elsevier—which resulted in the death of the journal. Among other things, such as journal property rights and the refusal to join the Initiative for Open Citation (I4OC) [4,5], the editors disputed Elsevier's "restrictive open access policies and prohibitive subscription costs" to publish with the journal using the open access (OA) option [6]. The publisher replied "that the current [Article Publishing Charge] is set at an appropriate rate ... lower than that of its main competitor and about average overall in the field, while offering above average content quality" [7]. It is worth mentioning that the publication fee (as of March 2019) for the JOI is set at 1800 USD (excluding taxes) which, ultimately, are charged in addition to the journal subscription fees. The literature has argued that the resources currently allocated in the subscription system of scholarly journals would correspond to a unit cost per article in the range between roughly 3800 and 5000 EUR (around 4300–5650 USD), which would make the large-scale transition towards OA a low-hanging fruit [8]. Nevertheless, the scenario might change over the next few years because, not only are the subscriptions becoming increasingly unaffordable, but OA publishing costs also face the problem of hyperinflation [9].

The economic aspect of the above conflict is quite fascinating because it has several implications, from the protest against Elsevier asking for reforms in its business practices [10]—which has recently gained heavy press coverage [11–13]—up to the ongoing OA transition and the shift towards open science [14–18], to mention just a few. It also implies the adoption of different, and perhaps contrasting, evaluation viewpoints. On the one hand, from a social welfare perspective, the former editorial board states that "Journals should serve the research community—not the other way around." On the other hand, the publisher defends its business model and the need to create value in favor of the shareholders.

Under the above framework, this paper aims to outline tentative answers to the following questions. Firstly, the business as usual (BAU) publishing charges are investigated from the market perspective, where BAU means that the financial return for the publisher should not be negatively affected if all the authors decide to adopt the OA model and subscription to ScienceDirect becomes worthless. The trade-off between subscription revenue and article processing fees mimics the method that has been proposed to support the transition to OA in hybrid journals [19] and avoid the phenomenon of "double dipping" [20]. Secondly, the analysis delves into how much Elsevier should charge for publishing all the articles under the OA model, assuming that the double-digit profit margin of the company reduces and does not exceed the market benchmark. It should be noted that the above hypothesis is to shrink only the revenue and profit margin; it does not affect the costs, the reduction of which represents an additional way to gain more equitable and affordable publishing fees [21,22]. Thirdly, using both the current publishing charges and the ordinary fees resulting from the previous step as a starting point, the study tries to assess how many articles would have to be accepted, in an OA-only publishing landscape, in order for the publisher to benefit from the same revenue and profit margin as in the recent past. In this paper, despite the fact that the OA publishing sector relies on several operational models [23,24], the expressions article processing charges and article publishing charges are used interchangeably (APCs from now on).

## 2. Methods, Data, and Assumptions

### 2.1. Preliminary Remarks and Analytical Model

The following information was used to answer the research questions posed above: financial figures made publicly available by the RELX Group, of which Elsevier is a subsidiary; the number of articles featured in ScienceDirect that are accessible by subscription; the publisher OA price list; and estimates of the ordinary profit margin for professional publishers. The data was processed using a three-stage model (Figure 1), where each stage corresponded to one of the research questions. According to the nomenclature and the symbols used in the diagram, the analysis could be formalized using the following equations:

$$APC = RGrev \times Ers \times SDss / SDpa \mid \overline{Epm} \tag{1}$$

$$oAPC = RGrev \times Ers \times SDss \times \beta / SDpa \mid Epm \simeq opm \tag{2}$$

$$naa = RGrev \times Ers \times SDss / APC \vee oAPC \mid \overline{Epm} \tag{3}$$

In other words, the BAU APC—Equation (1)—was calculated by dividing the product of the RELX Group revenue (*RGrev*), the Elsevier revenue share (*Ers*), and the ScienceDirect subscription share (*SDss*) by the number of ScienceDirect paywall articles (*SDpa*), under the constraint of keeping constant the profit margin of the publisher (*Epm*).

Subsequently, the ordinary APC (*oAPC*)—Equation (2)—was calculated by multiplying the RELX Group revenue (*RGrev*), the Elsevier revenue share (*Ers*), the ScienceDirect subscription share (*SDss*), and a reduction factor of the profit margin (β), and dividing the result by the number of ScienceDirect paywall articles (*SDpa*) so that the Elsevier profit margin (*Epm*) aligned with the ordinary profit margin for publishers (opm).

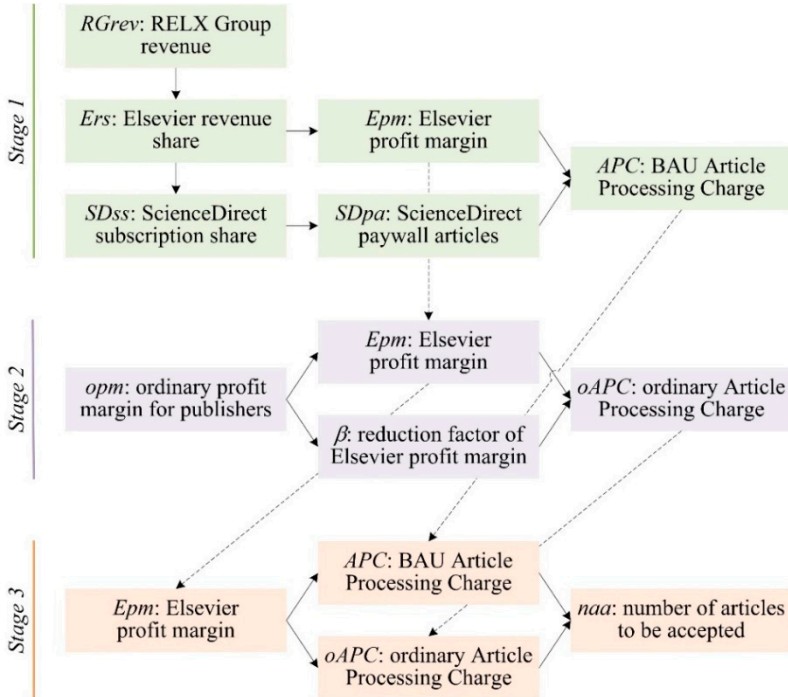

**Figure 1.** Three-stage analytical model.

Finally, the number of articles to be accepted (*naa*) in an OA-only publishing landscape—Equation (3)—was calculated by multiplying the RELX Group revenue (*RGrev*), the Elsevier revenue share (*Ers*), and the ScienceDirect subscription share (*SDss*), and dividing the result alternatively by the BAU *APC* and the ordinary APC (*oAPC*), under the constraint of keeping constant the profit margin of the publisher (*Epm*).

Because of the inherent uncertainty affecting the data, a stochastic approach was adopted here. For each uncertain input, a range of values and a probability distribution were estimated. Then, inputs were processed using a Monte Carlo simulation [25–27]. According to a very broad definition, under the umbrella of the Monte Carlo simulation is included "any technique making use of random numbers to solve a problem" [25] (p.1147). More to the point, the expression Monte Carlo simulation refers to a broad class of methods, essentially based on the use of stochastic (rather than deterministic) simulation algorithms, which include "some randomness in the underlying model" [26] (p.82), especially in the form of repeated random sampling. Below is a detailed description of the sources, a presentation of the data, and an insight into the assumptions required to perform the analysis.

### 2.2. Elsevier Financial Figures

According to Annual Reports and Financial Statements [28], the revenue of the RELX Group was 8385 million Euros in 2017, marking a growth of 4% over the previous year, while the adjusted operating profit scored 2604 million Euros, which is 31.1% of revenue.

The Scientific, Technical & Medical market segment represents 33.69% of the sales volume. RELX defines it as a "global information analytics business that helps institutions and professionals advance healthcare, open science, and improve performance for the benefit of humanity" [28] (p. 9, 14). It essentially corresponds to Elsevier and its scientific publishing activity, while other market segments are Risk and Business Analytics, Legal, and Exhibitions. Hence, Elsevier's revenue can be estimated to be around 2825 million Euros or (considering an exchange rate of 1.13 as at March 2019) 3192 million USD. Out of that revenue, 2% comes from advertising, 72% from subscriptions (2298 mUSD), and the remaining 26% is referred to as transactional (830 mUSD). It is assumed that the latter type includes, among other things, the APCs paid by authors and sponsoring institutions, so the analysis below

mainly focuses on subscriptions. The profit margin for Elsevier alone is up to 36.8%, considerably higher than that of the whole corporate group.

Annual Reports and Financial Statements for the year 2018 [29] provide the following figures (considering a GBP to USD exchange rate of 1.33 as at March 2019): Elsevier revenue is nearly 3376 mUSD, the subscription revenue share is 74% (2498 mUSD), and the profit margin is about 37.12%. The financial figures for the years 2017 and 2018 cannot be directly compared due to the adoption, in the latter year, of the new accounting standards IFRS 9, IFRS 15 and IFRS 16 [30].

A source of uncertainty lies in the fact that Elsevier sells several subscription-based services (e.g., Scopus, SciVal, ClinicalKey, and so forth), not only ScienceDirect. Building on other data [31], it is hypothesized that the share of revenue from ScienceDirect is 90% on average, varying in the range between 84% and 96% with a uniform probability distribution. The underlying assumption is that ScienceDirect provides the bulk of revenue the publisher benefits from.

*2.3. Subscription-Based Articles in ScienceDirect*

ScienceDirect provides access to nearly 716 thousand documents published in Elsevier journals during the year 2017. It is assumed that the cash flow generated by two article types (Conference abstracts and Conference info, about 115 thousand items) is a type of transactional revenue (under the form of fees paid to participate in the meetings), so those articles are excluded from the analysis. Other types of documents are also disregarded: Book reviews (less than 2 thousand), Editorials (more than 11 thousand), Errata (more than 3 thousand), and News (about 6 thousand) because they mostly refer to the editorial activity, and it is unlikely that authors are asked to pay a fee to publish them.

Two other document types are quite problematic. The item Encyclopedia includes about 5 thousand documents that are usually written by invitation, but they can also be autonomous contributions of the authors, who may be asked to pay for publication in an OA-only model. The item Correspondence features nearly 13 thousand papers; it is likely to include letters to the editor, which are not per se different from other peer-reviewed contributions of the authors, who again may be asked to pay in an OA-only landscape.

Therefore, several ranges are considered, where the lower bound includes fourteen article types (Review articles, Research articles, Book chapters, Case reports, Data articles, Discussion, Examinations, Mini reviews, Practice guidelines, Product reviews, Short communications, Software publications, Video articles, Other), while the upper bound includes two additional types (Encyclopedia, and Correspondence). The titles Procedia Engineering and Energy Procedia in 2017 and the title Materials Today: Proceedings in 2018 are excluded. Despite being indexed as Research articles, the documents they publish are, in fact, conference papers. ScienceDirect is likely to include other conference proceedings in the item Research articles, and that is an aspect that affects the reliability of the results.

In summary, the overall number of articles for the year 2017 vary between a minimum of 549,059 and a maximum of 567,358, of which 454,035–468,891 are accessible by subscription. The remaining 95,024–98,467 (nearly 17%) are OA documents. The data for the year 2018 are as follows: 600,350–618,805 articles; 530,479–547,160 paywall papers; 69,871–71,645 (about 12%) OA documents. Tables 1 and 2 summarize the key figures concerning Elsevier's market segment and ScienceDirect in 2017 and 2018, respectively.

**Table 1.** Summary of the key figures concerning Elsevier's market segment and ScienceDirect (2017).

| | Elsevier Market Segment: Revenue | Estimated ScienceDirect Revenue | | | | ScienceDirect: Articles [1] | |
|---|---|---|---|---|---|---|---|
| | mUSD | Pct. (min) | Pct. (max) | | | n. (min) [2] | n. (max) [3] |
| Subscription | 2298 | 84% | 96% | | Paywall | 454,035 | 468,891 |
| Transactional | 830 | | | | Open Access | 95,024 | 98,467 |
| Advertising | 64 | | | | | | |
| Total | 3192 | | | | | 549,059 | 567,358 |

[1] Source: author's study based on ScienceDirect data. Excluded items: Conference abstracts, Conference info, Book reviews, Editorials, Errata, and News. [2] The lower bound includes: Review articles, Research articles, Book chapters, Case reports, Data articles, Discussion, Examinations, Mini reviews, Practice guidelines, Product reviews, Short communications, Software publications, Video articles, Other. [3] The upper bound also includes: Encyclopedia, Correspondence.

**Table 2.** Summary of the key figures concerning Elsevier's market segment and ScienceDirect (2018).

| | Elsevier Market Segment: Revenue | Estimated ScienceDirect Revenue | | | | ScienceDirect: Articles [1] | |
|---|---|---|---|---|---|---|---|
| | mUSD | Pct. (min) | Pct. (max) | | | n. (min) [2] | n. (max) [3] |
| Subscription | 2498 | 84% | 96% | | Paywall | 530,479 | 547,160 |
| Transactional | 810 | | | | Open Access | 69,871 | 71,645 |
| Advertising | 68 | | | | | | |
| Total | 3376 | | | | | 600,350 | 618,805 |

[1] Source: author's study based on ScienceDirect data. Excluded items: Conference abstracts, Conference info, Book reviews, Editorials, Errata, and News. [2] The lower bound includes: Review articles, Research articles, Book chapters, Case reports, Data articles, Discussion, Examinations, Mini reviews, Practice guidelines, Product reviews, Short communications, Software publications, Video articles, Other. [3] The upper bound also includes: Encyclopedia, Correspondence.

## 2.4. Elsevier OA Price List

Elsevier provides a price list for publishing under the OA umbrella in 2197 of its journals (https://www.elsevier.com/__data/promis_misc/j.custom97.pdf, accessed on 13 March 2019), 1991 of which (about 91%) are hybrid, while the remaining 206 are fully AO journals. The average APC is 2544 USD, with a standard deviation of 828 (the few fees expressed in Euros are converted using the aforementioned currency exchange rate of 1.13). However, the frequency distribution is quite skewed (Figure 2). Actually, it somehow resembles a triangular distribution with a minimum value of 125 USD, a maximum value of 5200 USD, and a peak value of 3000 USD, so part of the analysis below rests on this distribution assumption. The figure of 3000 USD is recurring in hybrid journals—it was indeed adopted by Springer in its Open Choice program [32]—and is considered "more or less . . . a de facto industry standard" [33] (p. 920). Two additional remarks are in order here. Firstly, the price list does not include all the titles in "Elsevier's portfolio of 2500 journals" [29] (p. 14), and it is likely to exclude several tens of journals characterized by zero-APCs. Secondly, the mean value presented above is the simple average; the weighted average using the volume of articles published in 2018 as weight (according to ScienceDirect data) is, instead, 2824 USD, which is closer to the peak value of the distribution.

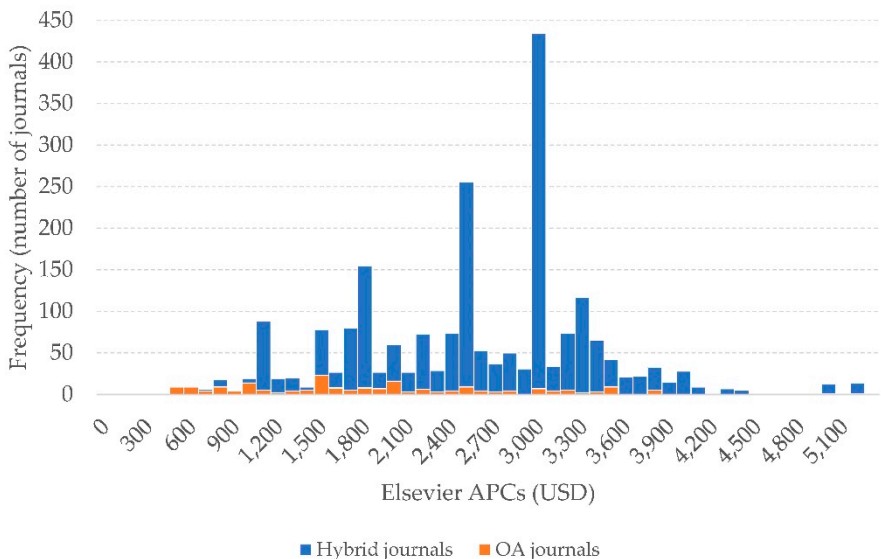

**Figure 2.** Frequency distribution of Elsevier's article processing charges (APCs) (as of March 2019).

### 2.5. Ordinary Profit Margin for Professional Publishers

Elsevier's double-digit profit margin—37.12% in 2018, 36.8% in 2017, in line with the previous years—has been recently highlighted and criticized in some magazine and newspaper articles [34–36]. However, a judgment on those figures is meaningless unless a comparison is made between profits and investment, as well as between Elsevier's performance and the market benchmark.

According to A. Damodaran (http://www.stern.nyu.edu/~{}adamodar/New_Home_Page/data.html, accessed on 14 March 2019), the Publishing and Newspapers sector is currently characterized by a gross margin of 39.11% in the US (33 listed companies, one of which is John Wiley and Sons Inc.) and 44.94% in Europe (85 listed companies, including Axel Springer SE). The profit margin—which can be deduced from Elsevier financial reporting by dividing the adjusted operating profit and revenue—is roughly comparable to the ratio between EBITDA—Earnings Before Interest, Taxes, Depreciation, and Amortization—and sales. It is estimated to be 10.94% and 12.61% for the Publishing and Newspapers industry in the US and Europe, respectively. Raw data published in magazine and newspaper articles point to an ordinary profit margin for professional publishers in the range from 10% to 13% [37,38], which confirms the above figures on the EBITDA-to-sales ratio. Therefore, Elsevier seems to outperform (by far) its competitors. What is more, the whole RELX Group shows a return on the invested capital of 13.1% in 2017 and 13.2% in 2018, which is near twice as much as the weighted average cost of capital of the Publishing and Newspapers sector, estimated to range from 7.26% in the US to 7.53% in Europe. Since the latter is a proxy of the expected average return of all the investors, RELX repays its investors with a return that is twice the benchmark figure. Hence, it is effective in transforming the near-term profit in a (huge) long-term value for the shareholders.

As per the above data, one can think Elsevier's profit margin could even be twice or three times lower. Here it is assumed that the ordinary profit margin varies between 10.0% (the lowest of the values suggested by the literature cited above, nearly three times less the current figure) and 18.5% (about half the current figure), following a uniform distribution.

### 2.6. Caveats, Implicit Assumptions, and Limitations

In the introductory section, it is stated that the transition to an OA-only model would make the subscription to ScienceDirect worthless. That assertion is only partly correct. Actually, the literature has the continuous ability to generate revenue and profits as, even in the future, already-published toll-access articles will be accessible by subscription or pay-per-view options. The above is indeed a

limitation of the analysis presented below, and the model used here is unfit to accommodate for this kind of issue.

The methodology adopted here also brings with itself a few additional, implicit assumptions which deserve mentioning. One of these is that—since BAU and ordinary refer to the financial return from the market point of view—for-profit publishers remain part of the process meant at sharing scientific knowledge, and even in the future will play a role in the dissemination of academic research.

Further hypotheses involve the agents and how they react to the changing publishing model. The authors, especially, should be prone to adapt their habits according to varying conditions so as to optimize both their costs (mainly the APCs) and benefits (for instance, getting published in prestigious and high-impact journals). Here instead, for simplicity's sake, they are assumed to exhibit unchanged behavior as far as the number and quality of the submitted papers are concerned.

On the publisher's side, the implicit assumption is that revenue and profits are mostly independent of its market power, which is likely to vary in the transition from a traditional publishing model, where libraries pay, to an OA-only model, where authors and sponsors bear the costs. Also, the third stage of the model is based on the premise that changes in the number of published articles will not affect the costs. Indeed, unless costs remain constant, the same revenue cannot result in the same profits. A further implicit, simplifying assumption is that APCs are essentially a function of financial drivers, regardless of the standing and prestige of its journals. That results in another limitation of the analysis below. Indeed, it is known that OA journals' impact plays a crucial role in shaping their APCs, while it is less important in regard to hybrid journals [39,40].

## 3. Results: APCs from the Market Perspective and Related Aspects

Under the hypothesis of a full transition to OA, the inputs processed using a Monte Carlo approach and one hundred thousand iterations (Tables 3 and 4) led to the following results (Tables 5 and 6).

**Table 3.** Summary of data and assumptions for the Monte Carlo simulation (2017).

| Item | Unit of Measure | Value | Distribution | Minimum | Maximum | Peak (Mode) |
|---|---|---|---|---|---|---|
| Revenue | mUSD | 3192 | | | | |
| Adjusted operating profit | mUSD | 1176 | | | | |
| Profit margin | Pct. | 36.8% | | | | |
| Subscription revenue | mUSD | 2298 | | | | |
| ScienceDirect share of subs. rev. | Pct. | | Uniform | 84.0% | 96.0% | |
| Paywall articles | n. | | Uniform | 454,035 | 468,891 | |
| Elsevier APCs | USD | | Triangular | 125 | 5200 | 3000 |
| Ordinary profit margin | Pct. | | Uniform | 10.0% | 18.5% | |

**Table 4.** Summary of data and assumptions for the Monte Carlo simulation (2018).

| Item | Unit of Measure | Value | Distribution | Minimum | Maximum | Peak (Mode) |
|---|---|---|---|---|---|---|
| Revenue | mUSD | 3376 | | | | |
| Adjusted operating profit | mUSD | 1253 | | | | |
| Profit margin | Pct. | 37.1% | | | | |
| Subscription revenue | mUSD | 2498 | | | | |
| ScienceDirect share of subs. rev. | Pct. | | Uniform | 84.0% | 96.0% | |
| Paywall articles | n. | | Uniform | 530,479 | 547,160 | |
| Elsevier APCs | USD | | Triangular | 125 | 5200 | 3000 |
| Ordinary profit margin | Pct. | | Uniform | 10.0% | 18.5% | |

Building on the 2017 data, the aim to preserve the profit margin of the publisher implies an estimated average APC of 4482 USD, far higher than the current mean APC (2824 USD being the weighted average using the volume of articles as weight, 2544 USD the simple average). In the first stage of the simulation, the stochastic variables—ScienceDirect's share of subscription revenue and the number of paywall articles—do not seem to affect the outcomes substantially. Elsevier's 2018 balance sheet data, jointly with the higher number of articles featured in ScienceDirect in the same year, leads to a downward shift of the results.

The estimated average APC in an OA-only model shows a value of 4173 USD; however, there is still a considerable gap with current APCs (2824 USD being the weighted average).

**Table 5.** Results: comparison of current and estimated average APCs (based on 2017 data).

| | Estimated Average APCs in an OA-Only Model (First Stage) | Estimated Average APCs in an OA-Only Model with Ordinary Profit Margin (Second Stage) |
|---|---|---|
| | USD | USD |
| Mean | 4482 | 3308 |
| Standard deviation [1] | 180 | 163 |
| Minimum | 4094 | 2883 |
| Maximum | 4892 | 3785 |
| Peak (mode) | 4475 | 3325 |
| Distribution | Normal | Normal |

[1] Standard deviation of the average APCs estimated in the 100,000 iterations of the Monte Carlo simulation.

**Table 6.** Results: comparison of current and estimated average APCs (based on 2018 data).

| | Estimated Average APCs in an OA-Only Model (First Stage) | Estimated Average APCs in an OA-Only Model with Ordinary Profit Margin (Second Stage) |
|---|---|---|
| | USD | USD |
| Mean | 4173 | 3066 |
| Standard deviation [1] | 166 | 150 |
| Minimum | 3826 | 2680 |
| Maximum | 4533 | 3486 |
| Peak (mode) | 4075 | 3075 |
| Distribution | Normal | Normal |

[1] Standard deviation of the average APCs estimated in the 100,000 iterations of the Monte Carlo simulation.

The scenario outlined in the second stage of the simulation is radically different (Tables 5 and 6). Building on the 2017 data, after dropping the constraint concerning the profit margin, the estimated average APC decreases to 3308 USD (+17.1% in comparison to the current weighted average APC of 2824 USD). Using the 2018 data, the estimated average APC in an OA-only model is 3066 USD (+8.6% in comparison to the current value).

The third simulation stage relies on the direct relationship that, in an OA environment, ties the publisher's revenue to quantity (number of published articles) and price (average unit APC). All else being equal, if APCs decrease, the revenue loss may be recovered by increasing the number of accepted articles, resulting in an unchanged profit if costs are constant. RELX Annual Reports state that "In the primary research market during 2017, 1.6m research papers were submitted to Elsevier ... resulting in the publication of over 430,000 articles" [28] (p. 14) and "Elsevier serves the global scientific research community, publishing over 470,000 articles in 2018 ... with 1.8m articles submitted" [29] (p. 14). Although the number of paywall articles analyzed here, based on ScienceDirect data, is several thousand higher (potentially because many conference papers are indexed as research articles), the above data imply an acceptance rate of about 26–27%. The analysis suggests that the rate could rise to about 35–45%, considering both the estimated ordinary APCs in an OA-only model (according to the second stage above) and current Elsevier APCs (according to the assumption of their triangular distribution). Furthermore, the optimal acceptance rate from the market perspective could be even higher and fall between 45–55% considering the simple average of current Elsevier APCs (Figures 3 and 4).

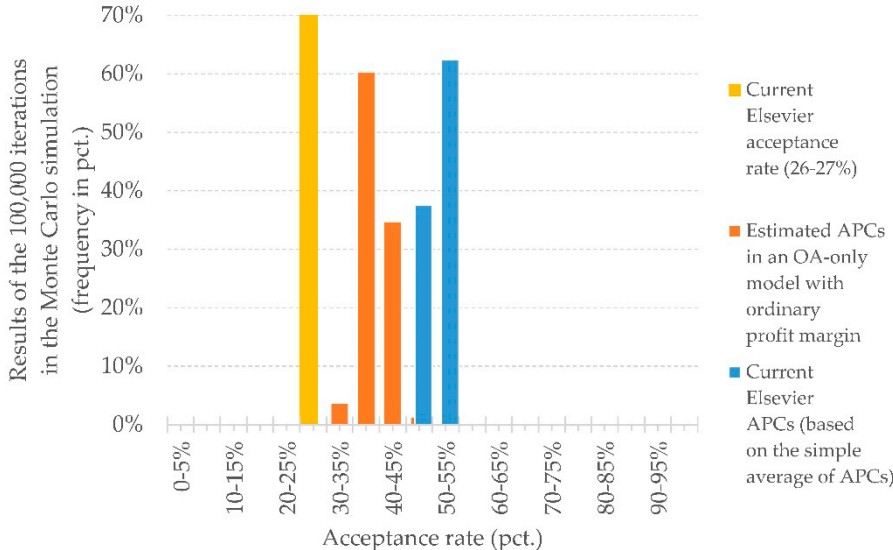

**Figure 3.** Results: potential acceptance rates with current Elsevier APCs in comparison to acceptance rates with estimated average APCs in an OA-only model and an ordinary profit margin (based on 2017 data).

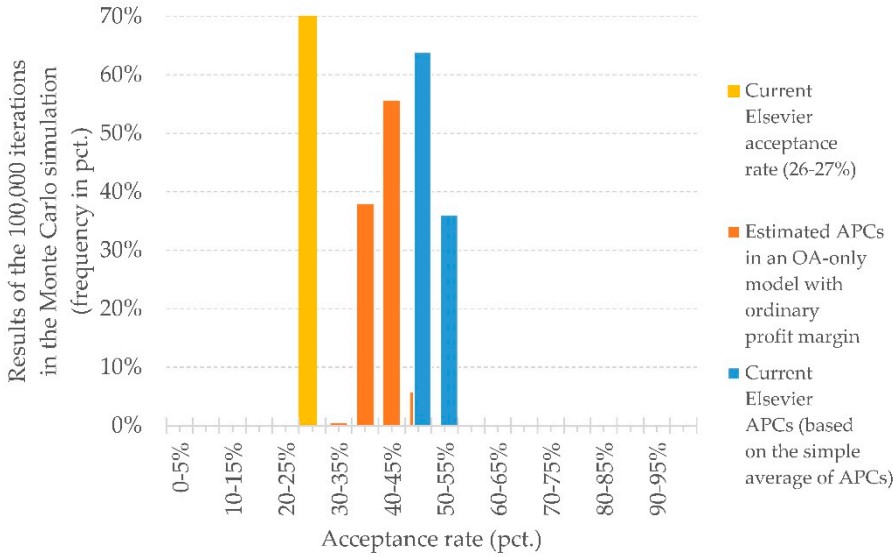

**Figure 4.** Results: potential acceptance rates with current Elsevier APCs in comparison to acceptance rates with estimated average APCs in an OA-only model and an ordinary profit margin (based on 2018 data).

## 4. Discussion: The Academic Publishing Business Model

It has been suggested that the dominant business approach in scholarly publishing, based on subscription selling, is becoming unaffordable and obsolete [41,42]. Although commercial publishers such as Elsevier believe "that the subscription model will remain popular among ... customers" [43] (p.98), it should be replaced, it has been claimed, by a mix of revenue sources, which is also essential to the long-term sustainability of the OA model [44]. Precisely the arising of the OA movement—reinvigorated by the growth OA publications have experienced so far [18,33,45–47]—has brought to the fore the question of who should pay for scientific publications, and whether it is legitimate or not that authors pay to get their works published. Regardless of the answers, it is long since known that, among the barriers toward the transition to OA, the business model plays a prominent role [48,49].

A long-standing debate focuses on the pricing level that would make a publishing venture financially sustainable [50]. Collections of data and investigations into OA APCs can be found in the literature throughout the last decade or so [51–56]. Far from being easily solved, the issue involves

the gap between the request for low APCs by OA advocates, the willingness to pay of authors and institutions, and the need of professional publishers to reach the breakeven point, namely, to cover costs.

The findings presented above suggest that, although commercial publishers are moving towards OA [57], that shift could lead to APCs far higher than the current level, especially in the case of for-profit professional publishers such as Elsevier. It is difficult to understand whether these findings can be generalized or not, inasmuch as the scientific publishing landscape is populated by different players [44,58]. Both university presses and non-profit scientific societies run journals, and also national academies of science play a role in the publishing community. On the one hand, one could expect that they differ from commercial publishers because they do not seek to maximize profits. Nonetheless, it is not unusual that learned societies benefit from double-digit surplus thanks to their publishing activity [59]. In addition, the purpose of professional actors should be better framed as the maximization of the (long-run) enterprise value rather than the (near-term) profit [60], with the implication that the care about the end-user and stakeholder need not differ between for-profit and non-profit publishing ventures. Furthermore, although publishers of different natures are prone to relying on distinct revenue sources, some case studies call into question the assertion that OA and traditional subscription-based journals do not share the same business model [61]. Broadly speaking, whatever the target is, non-profit players also need to cover their costs, which is likely to happen for APCs at a level not too different from that which we are experiencing today, unless substantial savings can be achieved in the investment and operational costs [21,22,62].

OA is not only a matter of social justice [63,64] but also tends to maximize social welfare [65,66], in accordance with the view that scientific information is an economic public good [67] and brings about positive externalities [68]. Nonetheless, the literature has already expressed the concern that moving from the reader-pays model to a landscape where authors (or sponsors) bear the costs could impair the quality level of journals, while subscription fees act as an incentive to preserve a high level of content quality. This is because for-profit professional publishers are likely to perceive an economic incentive to increase the article acceptance rate [65], hence, they may be willing "to publish low-quality articles in order to increase [their] profit from author fees" (p. 224). Regardless of the phenomenological expression of the issue that has been dubbed "predatory publishing" [69]—in the strict sense of the term, namely, deceptive publishing practices—the references cited above are mostly concerned with the potential downward turn in quality that may affect legitimate journals and well-established publishers. The issue has been addressed in several papers [70–72]. The findings presented above concerning the potential increase of the acceptance rate are in line with that school of thought. The same topic, from different analytical perspectives, is mentioned in other studies [73,74], although the authors that publish using the OA option do not seem to perceive it as a matter of concern [75]. Nevertheless, it should be stressed that there are also counteracting forces, which incentivize editors and publishers to maintain high standards for their journals, as is the case of the ongoing pressure to publish in high-impact venues. For instance, as long as scholars rely on journal impact measures in order to boost their career perspectives [76], OA publishers can benefit from higher APCs by improving the standing of the journals they run. Furthermore, it can be questioned whether highly selective journals are still required in the digital publishing era, as well as whether selectivity, rather than openness and accessibility, is their most appreciable characteristic [77].

## 5. Conclusions

This study focuses on three issues of the transition towards a fully OA publishing landscape and analyzes them from the perspective of a big commercial publisher, namely Elsevier. The first issue concerns the APCs level in the case all the articles are published using the author-pays model, under the hypothesis of a constant profit margin in comparison with the recent past. The second issue is how much should the publisher charge to publish all the articles under the OA umbrella, assuming it benefits from an ordinary profit margin. Finally, the third issue is framed as follows: how many articles would have to be accepted, in an OA-only publishing landscape, so that the publisher benefits from

the same revenue and profits as in recent years. The results point at APCs that, on average, would be around twice the current level in order to preserve the publisher's profit margin (4173–4482 USD in comparison to 2544 USD). By relaxing that constraint, a downward shift of the APCs can be expected so they would tend to get close to the current peak value (3066–3308 USD). For comparison's sake, it deserves mentioning that the transformative OA agreement known as Projekt DEAL, which the German Rectors' Conference has already signed with Wiley and is going to sign with Springer Nature, is based on an APC of 2750 Euros (about 3100 USD), in addition to 2 million Euros as a lump-sum access fee. Finally, the article acceptance rate would be likely to grow from 26–27% to 35–55%.

The analysis is based on multiple explicit and implicit hypotheses. It also suffers from several limitations, especially the inherent uncertainty of many inputs—which has been dealt with using a Monte Carlo simulation approach—and the fact that several implicit assumptions turn out to be an oversimplification of the publishing environment. Furthermore, the analysis includes both hybrid and fully OA journals, without distinguishing between them, and regardless of the fact that they can be subject to different transformative agreements in the transition towards the OA model. Nevertheless, the results may contribute to advancing the debate on that transition, at least as far as its financial viability from the perspective of commercial publishers is concerned.

**Funding:** This research receives no external funds.

**Conflicts of Interest:** The authors declare no conflict of interest.

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
