# Peer review of "Business as Usual with Article Processing Charges in the Transition towards OA Publishing: A Case Study Based on Elsevier"

_publications, doi:10.3390/publications8010003_

Round 1
Reviewer 1 Report
The article is very interesting especially in these days in which Plan S is a hot topic and it will likely the publishers business. Said that, I have nothing to say about the content, but the title. It seems very generic when the author analyses only the case of Elsevier. My suggestion is to modify the tiltle accordingly to reflect the content. My suggestion:
"An approach from pay to read to pay for publishing. Elsevier case study" or something similar, obviously the author has the last word
Author Response
Please, find the detailed responses to all the reviewer's comments in the attached file.

Reviewer 2 Report
Brief summary:
The paper analyses how a total shift from closed-access to open-access publishing could affect the big, traditional STM publisher “Elsevier”. Three scenarios are calculated. First, how high should be the APCs for articles in Elsevier’s journals to preserve Elsevier’s revenue and profit margin? Second, how high would be the APCs if Elsevier generates just a (slightly above) market-average profit margin with unchanged costs? The first two scenarios assume an unchanged number of published articles. Third, how much articles should Elsevier publish to maintain its current profit margin (and revenue) charging either current APCs or the APCs from the second scenario. The results point to (1) an average APC between 4,000 and 5,000 USD, (2) an average APC between 3,000 and 3,300 USD, and (3) a considerably increase in the acceptance rate from 26% to 40%, or even 50%. Although not intended by the author, the calculations show how much libraries and other subscribers of Elsevier journals globally pay each year for access to each newly published article in the current subscription-based system.
Broad comments:
The author looks from the market’s viewpoint on the strategic options a big publisher has to encounter the open-access transformation of scientific literature. Consequently, the question is not “how much would [it] cost to publish all the articles under the OA model”, as asked in the abstract (line 15/16), but how much should Elsevier charge for publishing all the articles under the OA model. That Elsevier could be tempted to charge on average 4,000 to 5,000 USD to preserve its revenue, does not automatically mean that authors will indeed pay these APCs. They could choose to submit their articles to journals with comparable scope, quality and reputation of other publishers (e.g. Springer Nature, Wiley etc.) with lower APCs. Thus, some of the scenarios can become unfeasible for Elsevier. Throughout the paper, the author should check the perspective (from Elsevier’s viewpoint, not the viewpoint of publishing scientist, see e.g. line 50/51) and keep in mind that it is not only up to Elsevier to determine which scenarios will realize. This is particularly important for the conclusion.
At the beginning of the introduction, the author introduces the topic with a reference to the recent dispute between the Elsevier management and the editorial board of the Journal of Informetrics. However, the author cites the editors incorrectly. Actually, the editors criticize Elsevier’s “restrictive open access policies and prohibitive subscription costs” (http://www.issi-society.org/blog/posts/2019/january/resignation-of-the-editorial-board-of-the-journal-of-informetrics/ ). My interpretation of the editorial resignation letter is that they point at Elsevier’s open-access policy that does not allow authors to publish their accepted manuscripts on a repository without an embargo period using the green route of open access. Moreover, the publisher could conclude transformative contracts with libraries that would allow authors to publish open access in Elsevier-hybrid journals without additional costs. However, Elsevier stands out from other big STM publishers as it refuses transformative agreements that do not increase its revenues (the only exception is the recent contract with Norwegian institutions). The former JOI editors did not complain that the APC of 1,800 USD is too high, but that it is charged on top to the prohibitive subscription costs. In principle, APCs could be zero within transformative agreements (Read-an-Publish) without impairing the publisher’ revenues and profit margin. In my interpretation, this is meant by the claim of the editorial board that the “the amount of Article Publishing Charges (APCs) of JOI are non-negotiable for Elsevier” (http://www.issi-society.org/media/1380/resignation_final.pdf). To conclude, the introduction in the whole topic rests on an incorrect citation. It comes as little surprise that this results in a flawed conclusion.
The author calculates the average APC according to Elsevier’s price list. Actually, the price list includes about 200 zero-APCs, so the minimum APC is not 125 (line 145) but 0. Maybe the author excluded APCs for OA journals, which should be mentioned. Moreover, there is no indication that the calculated average is weighted for article output. Elsevier might have high-volume journals with relatively low APCs, so that the article-weighted average APC could be lower. This is important when it comes to the comparison between the current average APC and estimated average APCs (scenario 2 and 3). The later are effectively article-weighted.
The author mentions four implicit assumptions (lines 177–192). The last one is that APCs are not a function of the standing and prestige of the journals. However, a recent paper finds that the journal’s impact is one of the most important factors for the level of APCs. The journal’s impact is crucial for the level of APCs in open-access journals, whereas it little alters APCs for publications in hybrid-journals. (Schönfelder, N. APCs—Mirroring the impact factor or legacy of the subscription-based model?, 2018. https://doi.org/10.4119/UNIBI/2931061). Moreover, the author should not take the implicit assumptions at face value when he derives conclusions for the open-access transformation in general.
The results for the first question/scenario regarding the standard deviation look strange. I do not understand why the standard deviations of the estimated APCs are many times lower than the standard deviation of the current APCs. In fact, they should be higher as the uncertainty concerning the number of paywall articles, the Science Direct share of subscription revenues, and the ordinary profit margin introduces more variance. I suppose, the author did not estimate the APCs in an OA-only model but estimated the average APC in an OA-model. Hence, the standard deviation in Table 5 third column (first stage) is the standard deviation of the estimated, average APC, not the standard deviation of the estimated APCs. This is a tremendous difference and important for the third scenario.
The author mentions a caveat that applies to the second and third scenario (lines 223–225). It is the question whether authors and their institutions can afford the estimated APCs (“the financial burden of a fully OA publishing model”). Globally, the answer is definitively yes, as the author’s calculations impressively show. All scenarios assume that research and higher-education institutions will pay at least the same amount to Elsevier as they currently do, leaving Elsevier’s revenues unchanged.
In the third scenario, the author should make an additional assumption explicit (e.g. in line 229). It is assumed that increasing the number of published articles does not increase the publisher’s costs. I would favor to reshape the question, as e.g. “The third issue is how many articles would have to be accepted, in an OA-only publishing landscape, so that the publisher benefits from the same revenue as in the recent past.” This is what the author actually calculates. He divides the current revenue by the average APC. The same revenue can only result in the same profit if the costs do not change.
I wonder about the variance in the estimated acceptance rates in the Figures 3 and 4. The variance of the estimated acceptance rates based on Elsevier’s current APCs are much higher than the variance based on the estimated average APCs. I suppose, the author used the standard deviation of the current APCs and the standard deviation of the estimated average APC to simulate the acceptance rates. However, this would be wrong. The current average APC at Elsevier is deterministic not stochastic. To say it in another way, its variance is zero. The simulated acceptance rate should be between 42% and 54% for 2017 (if calculated with the unweighted average of current APCs).
The expressed concern that the APC business model could impair the quality of journals is well documented in the literature and the outcomes are called “predatory journals”. But also publishers of subscription-based journals have economic incentives to increase the article volume either via issuing new journals or via accepting more articles. Especially the first route in combination with selling journal bundles to libraries is popular. By this, publishers can raise their revenues and increase their market share. In contrast, publishers of open-access journals have some economic incentives to run highly selective journals. As long as researchers rely on journal impact measures for their career promotion, publishers can charge higher APCs for high-impact journals. A publisher can optimize its revenue by the number of published articles and the level of APCs. Moreover, there are some genuine open-access publishers that were successful in developing above-average to high impact journals (e.g., Copernicus, PLoS, and Frontiers).
Rather than deriving conclusions from the analysis, the author expresses his personal opinion towards the open-access transformation. In line 293–295, the author starts to argue via the publisher’s costs—something that has not been analyzed before. Maybe the author mixed up the publisher’s and the scientists’ perspective. Moreover, the statement that the APCs are a reliable proxy of the appropriate cost level for such kind of venture, and that they are not upward bias, seems to be ridiculous as Elsevier’s profit margin is twice the benchmark figure. The last sentence unmasks the author’s concern not to have high-impact journals for scientists to publish in an OA-only world, which would threaten career promotion as long a research evaluation is based on journal citation-impact measures, rather than on the research’s own achievements (which could partly be measured by citations on article level, see, for example, the San Francisco Declaration on Research Assessment (https://sfdora.org/read/) and the Leiden Manifesto (http://www.leidenmanifesto.org/)). I gained the impression that the author uses the term “quality” as a synonym for “high impact”. Indeed, it is an open question whether highly selective journals are needed at all in the digital era, and whether selectivity is in fact a “primary need” (see Armstrong, M. Opening Access to Research. The Economic Journal 2015, 125, F1-F30, doi:10.1111/ecoj.12254). To enhance scientific communication, journals should publish technically sound articles rather than selecting for perceived importance. Readers can evaluate themselves to what extend the particular article enhances the scientific knowledge.
To conclude, I would suggest to
address the methodological issues,
not use the case of JOI as an introductory example or, at least, re-write it,
check the perspective (Elsevier vs. scientists) everywhere in the paper,
derive the conclusions from the analysis and hallmark personal opinions.
Specific comments:
Some sentences are hard to understand because of grammatical mistakes, e.g. line 15 and 50 (subject is missing) and line 250–253.
Misleading use of the term “incidentally” (line 33, 87, 93). I suggest dropping it.
A reference for the number of published articles is missing in Tables 1 and 2.
Re-label the y-axis in the Figures 3 and 4 to be more meaningful.
Author Response

(The authors gave the same response as above.)

Reviewer 3 Report
There are some excellent points made in the paper. However, the complexity of the issue, exacerbated by the writing style makes reading very difficult.
The title of the article should be changed as the article deals with Elsevier's subscription-based journals only (with hybrid articles in the journals).
The author needs to explain the statistics in a much simpler manner to allow the reader to follow the argument. The flaw with the statistics is that the roll-over profit through subscriptions is much more than has been factored into the manuscript. There is continued capacity for the article to generate income for the long term (not just per year) through subscription and 'pay-per-view'.
The author is strongly encouraged to stay away from casual commentary, such as "I wonder", "I suppose" and "I assume", as this is a scholarly publication and has to adhere to to the academic rigour associated with the publication.
Equitable access should be couched as social justice, and not social welfare.

Author Response

(The authors gave the same response as above.)

Round 2
Reviewer 2 Report
2nd Review of “The issue of business as usual and ordinary Article Processing Charges in the transition towards the author-pays model: A case study based on Elsevier Open Access price list” by Sergio Copiello
Brief summary:
The paper improved greatly since the last review. It is now focused on the publisher’s perspective and the conclusion are derived solely from the provided analysis. Still, there are some methodological issues, which I would recommend to address before publication.
Comments:
Elsevier’s APC price list: Indeed, Elsevier does not list zero-APCs in its up-to-date price list. However, in older versions (e.g. Elsevier OA Price List from 2018-10-08, see attachment and maybe accessible via the Internet archive “Wayback Machine” https://archive.org/) they are included. For example, I checked the journal “Acta Pharmaceutica Sinica B”. This journal has still zero-APC, but is not included in the up-to-date price list, as potentially about 200 other Elsevier journals. So at this point, I completely agree with the current manuscript and no further changes are necessary.
Weighted vs. simple average APCs: I appreciate that the author calculates and reports the current article-weighted average APC for Elsevier journals (although the data source for the weights remains unclear). This enables the reader to compare the estimated average APC with both benchmarks. However, I disagree that the estimated average APCs resemble the simple average rather than the weighted average. In Equation 1 and 2, revenue is divided by the number of articles. Revenue in the APC business model is the sum of APCs multiplied by the number of published articles for each journal. This is correspond exactly to the calculation of the weighted APC. To say it in other words: If you multiply the weighted average APC with the number of published articles, you will get the exact revenue whereas multiplying the simple average APC with the number of published articles could lead to a different figure.
Estimated APCs: I am pleased to see that my educated guess concerning the estimated APCs is correct: Actually, the estimated APCs are estimated average APCs. Therefore, I have to accentuate the need for changes in the presentation of the results.
First, changes concerning some terms are needed throughout the text, tables and figures. Instead of “estimated APCs”, the author should write “estimated average APCs” everywhere. When the current average APC of Elsevier is used, it should be labelled appropriately, see figures 3 and 4, light blue bars.
Second, estimated averages should always be compared with the current average, not with the peak (modal), minimum or maximum of current APCs. See, e.g., line 235, there the modal value of the estimated average APCs is compared with the modal value of current APCs (similarly in lines 244, 251/252, 253/254). This is highly misleading. A reader with less proficiency in statistics may not notice the difference. It is like comparing apples and oranges. Moreover, I would suggest to delete completely the first column of the tables 5 and 6 (current Elsevier APCs) and, instead, recall the current weighted average APC in the text.
Third, the simulated acceptance rates based on the current Elsevier’s APCs with the triangular distribution (dark blue bars) should be omitted as the comparison with the simulated acceptance rates based on the average APCs is totally misleading (see discussion above). If the author intends to demonstrate what the acceptance rates would need to be in case Elsevier charges on average APCs of 500, 1000, 1500 USD and so forth, it is better proceed as follows. Assume a uniform distribution from 1 to 10,000 USD, run the simulations with such a distribution as input and report the results with correct labelling in Figures 5 and 6 (rather in the background).
Finally, I would suggest inserting the current acceptance rate (26-27%) in the Figures 5 and 6 as a benchmark. This would enhance the comprehensibility of the figures.

Author Response

(The authors gave the same response as above.)

Reviewer 3 Report
Whilst a better second draft has been submitted, further proof-reading is still very much needed. Also, there are references in the text, which should only be in the reference list.
A rationale as to the choosing of the Monte Carlo simulation is still not provided. Even though this is an OA-focussed paper from an economics perspective (APCs and profit), readers who look at the title and abstract are not going to understand this simulation. Further, a very brief definition is needed (even if it's a footnote) of this simulation.
Please provide evidence from the literature to substantiate the phrases in line 197, "suggested by the literature" and line 223, "it is well-known that OA journals".
The sentence is not complete in line 225.
My strong recommendation is that you do not bring in "predatory publishing" - line 317 - so late into your article. It is not needed as the focus of your article is on the economics behind Elsevier's APCs. Bringing in predatory publishing when it has different meanings for the global north and global south (Raju, 2018), deflects the reader's focus, rather than focussing on the topic at hand.
Author Response

(The authors gave the same response as above.)
